# Classification of Key Elements of Construction Project Complexity from the Contractor Perspective

**Marin Nikolić [1,*]** and **Anita Cerić [2]**

1 Faculty of Civil Engineering, University of Zagreb, 10000 Zagreb, Croatia
2 Department for Organization, Technology and Management, Faculty of Civil Engineering, University of Zagreb, 10000 Zagreb, Croatia; anita@grad.hr
* Correspondence: marinnik9@gmail.com

**Abstract:** Contractors are facing an increasing degree of complexity in their construction projects. Due to inadequately prepared project plans, they have been suffering significant losses during the execution of construction projects. One of the key disadvantages of such plans is that during the planning process, a construction project is mostly defined as a linear rather than a dynamic and complex process with a high degree of uncertainty. Therefore, a contractor who is in the planning phase of a construction project should consider the impact of the project characteristics on its implementation according to the elements of project complexity. In this research, we therefore first made an overview of the existing research related to the elements of project complexity. Based on the frequency of their occurrence in existing surveys, this paper singled out eight groups of complexity characteristics that contractors should be aware of during construction projects. After that, based on the frequency of occurrence in the existing surveys, fifteen elements of complexity were classified for each project complexity group. The research conducted among construction project managers identified key complexity elements of the construction project from the contractor's perspective. Thereby, the classification of groups with the associated key elements determining the complexity of a construction project from the perspective of the contractor was performed. By properly analyzing the impact of key elements of complexity on project flow during the planning phase, contractors can be more successful when planning the project objectives to be performed.

**Keywords:** construction project; project complexity; construction project complexity; project management; project success

## 1. Introduction

The success of contractor organizations in the construction sector, which mostly operates as a project-oriented, depends on the success of individual projects. In the case of construction projects, success is measured on the basis of the results achieving the project objectives related to cost, time, quality, safety, and environmental conservation [1]. Understanding and properly dealing with project complexity is the key determinant of success, especially in project-oriented organizations [2]. The literature dealing with complexity defines uncertainty as one of its essential determinants [3,4], and, going along with it, the risks of the project and the consequences that they can cause. Successful completion of a project depends significantly on the number of activated risks in the project that cause deviations from the planned results and project goals, which are set in advance.

Problems during projects that cause cost and deadline overruns have been a common hurdle that researchers have dealt with for years [5]. Among the reasons why projects fail are their increasing complexity [3] or underestimations of the complexity of the project during the planning phase [6]. In practice, the contractor for the construction project during the planning phase, while defining the budget and time required to execute the project, views the project as being a proper and predictable process. However, a more

detailed analysis of practical examples leads to the conclusion that construction projects are non-linear and dynamic, i.e., complex processes that often exist on the border of chaos. Therefore, in order to influence the results and success of the project, the contractor should consider the project as a complex and unpredictable process when developing the plan.

One feature of unsuccessful projects is a delayed response to a problem that occurs during the execution of the project, while a feature of successful projects is the prediction of such problems [7]. Each project that the contractor performs has different variable and immutable characteristics and requires a different approach to management and execution. When planning a construction project with a contractor, it is necessary to analyze and determine the project's degree of complexity. The degree of project complexity is an indicator of the impact of the project characteristics on the activation of the risk in the project and therefore the impact of these complexity characteristics on the goals and success of the project as well. To determine the degree of project complexity, it is first necessary to define the term complexity of the project from a contractor perspective and the elements that affect its degree.

Although project complexity is not clearly defined [3,8–11], it is recognized as one of the critical characteristics of a project that determines the appropriate actions that will result in the success of the project [12]. It is widely accepted that the complexity of the project affects the results of execution and possibly results in the success of the project [13–16]. Baccarini [12] stated that project complexity helps to determine the requirements in terms of the planning, coordination, and control of the project; makes it difficult to clearly identify and define the objectives of the project; and plays a major role in selecting the appropriate organizational structure, selecting project inputs, and selecting the appropriate procurement arrangement for the project. As a determinant of the project, complexity significantly affects the project objectives related to time, costs, and quality. Based on the need to create a realistic plan for the execution of the project with indicators through which the successfulness of the project is monitored, Wood and Gidado [17] found that the degree of project complexity should be identified at the earliest stage of the project.

Measuring the complexity of construction projects is different for investors, designers, contractors, project managers, and site managers [18]. In previous surveys determining the complexity of a construction project, the elements that make up complexity, and how to measure it, the research was generally carried out for projects, IT projects, and construction projects. These surveys were mainly conducted from the point of view of investors.

In their work [11], Luo et al. analyzed the previous research related to the complexity of construction projects in detail and state in their conclusions that future research should address elements of project complexity from the perspectives of different participants in the construction project, the connection between complexity and project success, and how to increase complexity over the course of the project. Although research related to the complexity of construction projects has intensified over the last twenty years, a review of the literature found that there are no significant complexity studies that focus on the perspectives of contractors. As investors are increasingly using forms of contracting that transfer risks to the contractor, contractors encounter a greater degree of project complexity than investors [19].

The main objective of this paper is to identify groups of complexity and the key elements of the complexity in construction projects from the perspective of the contractor. Project complexity groups were determined based on a review of the literature and the frequency of occurrence in existing surveys and complexity models. Key elements of complexity were identified based on the literature review, as well as their frequency of occurrence in existing surveys and research conducted among construction project managers with contractors. In this way, a classification of groups of complexity and the key elements of project complexity is obtained and can be viewed from the perspective of the contractor. This paper presents the literature related to complexity, with special reference to the definition of project complexity and the complexity of construction projects. After that, the elements of project complexity are identified and summarized from the perspective

of the contractor, and the research methodology is presented. The research question for project managers which was included in investigation on project complexity was: Which element of complexity do you consider most important from the contractor's perspective? When we ask how managers perceive project complexity elements, we do not ask only in a specific case of a given project but rather in a more general manner. In other words, the question concerns the mental model, not the assessment in a single case. The results of the research show the degrees of importance of key elements of complexity within the different groups of complexity. Based on the results of the research, a classification of the complexity groups with the associated key elements of complexity was determined from the perspective of the contractor on the construction project. A discussion and guidelines for future research are provided, along with conclusions.

## 2. Literature Review

### 2.1. Project Complexity Theory

Project Complexity Management is becoming an important segment of project management that is crucial for the success of projects [20,21]. It is important to understand the link between project success and different project conditions, and it represents an intervention variable for successful project implementation [22,23].

The beginnings of the application of complexity theory in project management are related to works by Morris [24], Bennet and Fine [25], Bubshait and Selen [26], Bennet and Cropper [27], Gidado [28], Wozniak [29], and Baccarini [12]. All of these studies were mainly devoted to illuminating the notion of complexity in general, both in terms of projects and its impact on project objectives, though to a lesser extent.

When talking about complexity, its unpredictability (the degree that needs to be determined) depends on the interdependence and dynamic interaction between the system and individual elements of the system itself. In order to prevent a project's unpredictability from growing and creating additional problems, it is necessary to define the degree of complexity as soon as possible. In this way, the possibility of unpredictability (risk activation) is reduced, and the chances of the project being successful are increased.

From the point of view of complexity theory, each project is complex [30]. The claim that we often hear from those participating in different projects is that their project is complex from the point of view of complexity theory, but this is not a clear determinant. All of the authors above stated that the participants believe that their projects have a high degree of complexity and that because of insufficient knowledge of this theory, they reach for such definitions. Thus, every project, even the smallest, has complexity as its basic determinant; the degree of this complexity changes depending on its characteristics. The idea is that project complexity is applicable to all projects, regardless of their size.

Processing the complexity of the project is one of the most important but also the most controversial topics in project management. It is considered controversial because most standards, as well as researchers in the field of project management, have a different view on project complexity. Through a large number of surveys, there is still no consensus on how to define an unambiguous concept and measurement system to determine project complexity [31]. In the last few decades, complexity theory has been used in various fields, such as physics, astronomy, finance, biology, geology, chemistry, and meteorology [32,33].

### 2.2. Definition of the Project Complexity

The problems with complexity begin with the very notion of complexity [34]. Given that this is a multidimensional concept, providing an unambiguous definition of project complexity is still impossible. The most frequently quoted researchers are more concerned with the question of what complex systems are not than with what they actually are. In general, complexity refers to the difficulty of understanding certain phenomena in a given context or environment. In more specific terms, its use denotes a complex interaction between parts of a system. Complexity is defined in different ways across different groups of disciplines and in relation to different systems. However, as already stated, there is still

no consensus on the exact definition of complexity [33,35–43]. A detailed review of the literature finds a large number of definitions of complexity. Sinha et al. [44] found that there is not a single concept of complexity that can adequately encompass the intuitive notion of what complexity should mean. The first definitions of complexity were encountered in the 1950s and 1960s. However, it was only over the past thirty years that the number of authors studying what constitutes the notion of complexity increased. Baccarini [12] created one of the foundational works related to project complexity and found that overall project complexity does not exist and that we need to determine different types of complexity when we talk about project complexity. Complexity theory provides a general definition of complex systems for certain fields and analyzes the interaction of individual elements of complexity within those fields (e.g., the financial market field, IT sector field, construction field, biology field, etc.).

By reviewing complexity research, it can be established that it deals with elements that affect the degree of complexity, the impact of complexity on the project, and methods for measuring and managing complexity during the project [45,46]. A significant number of surveys address the elements of complexity as determinants of complexity and show the disaggregated structures of the notion of complexity itself. Complexity elements are arranged into groups of complexities, while individual authors determine the elements of complexity as independent indicators of the overall complexity of the project.

### 2.3. Complexity of the Construction Project

The notion of complexity is often used when talking about construction projects. Construction projects consist of a large number of elements, and their implementation requires a large number of participants and a large number of resources, as well as various techniques for their management. With their characteristics, construction projects match our general understanding of something that is significantly complex.

Project complexity is one of several concepts that represent the irregular behaviour of the project, but in the field of construction, this concept is of utmost importance [47,48]. As a scientific discipline, complexity is an emerging topic, but it is also a critical topic in the field of construction project management [21]. Construction projects are immutably complex and have been progressively becoming more complex since World War II [12]. In fact, construction processes can be considered to be some of the most complex ventures across all industries [49]. Today, the construction industry has experienced speedy progress in projects of rising size and complexity [43]. Large scales, sophisticated technical processes, long lead times, huge numbers of people involved, diverse geographic locations and high-performance pressures make these projects more complex than ever [50].

Although complexity is a widespread term that can be associated with any subject, there is still a lack of published literature in the field of complexity in construction. Thus, construction projects are often described as complex, but there is no universally accepted definition of complexity in the construction industry [17]. When it comes to complexity, it is most often analyzed from the theoretical or abstract perspectives, while the practical applications of complexity theory are very small. A large number of authors have noted the difficulty of applying theory in practice in relation to complexity theory [51–55].

The construction industry has shown great difficulty in dealing with the increased complexity of major construction projects [56]. Bertelsen [38] explained that the general view of construction projects is such that they are considered to be a regulated and linear phenomenon that can be organized, planned, and managed in a proper manner. Frequent examples of failure to complete construction projects on time and within the established plans has led to a reconsideration of how accurate such a general view of construction is and has forced us to consider whether construction projects are as predictable as we consider them to be. A more detailed processing of examples from practice led to the conclusion that construction projects are essentially non-linear, complex, and dynamic processes that often exist on the border of chaos. Therefore, he concluded that the perception that projects are regular and linear by nature is a crucial mistake that could impair the success of the project,

and project management must consider the project as a complex and dynamic process with non-linear characteristics.

Continuous requirements related to the speed of construction; cost and quality control; workplace safety and dispute avoidance, as well as the technological progress; economic liberalization and globalization; environmental issues; and fragmentation of the construction industry have led to a significant increase in the complexity of construction processes. Regardless of how their relationship was conceptualized in the literature, it is clear that complexity and uncertainty have substantial impacts on project performance. Today, complexity has reached a level where construction project managers have to consider its impact on the success of the project with great care [28]. It is a common opinion that the reason for poor results lies in the planning process and that construction processes are significantly more complex for a large number of reasons [12,57,58]. Most believe that the main reason for the failure of a construction project is poor project management. The critics are partly right. However, they are correct not in the way that they think they are. In order to properly manage a project, it is important to understand the nature of the project. From the above, it can be concluded that we do not know enough about the nature of the construction projects that project managers try to manage in the right way.

The complexity of construction projects significantly influences various aspects of the project results [18]. Many empirical studies in the field of construction have found that complexity affects project duration, cost, and quality [43,59–65]. It is widely accepted that the complexity of a project should be something that can be measured objectively for the purposes of continuous feedback, which would help control the project implementation process [12,66–70]. A systemic analysis of the complexity of construction projects is a crucial element in decision making used by project managers and in the successful implementation of complex construction projects [11]. Complexity is crucial for project resource management, and finding ways to manage complexity can affect the improvement of project results [70]. Leung [70] stated that it is necessary to define a quantitative method for measuring the complexity of construction projects.

### 2.4. Project Complexity from Contractor Perspective

Contractors are the main actors concerned with project management performance and need to manage their perceived project complexity [71]. When we talk about the complexity of a construction project, it is important to emphasize that it can be viewed from the different perspectives of the participants involved in the construction project. A significant degree of project complexity for the investor does not necessarily mean the same or similar degree of project complexity for other participants who are involved in its execution. It is necessary to specifically analyze the complexity for each of the participants in the construction project from their individual perspective. Previous research has mainly dealt with the analysis of complexity from the perspective of investors, but in accordance with the research of Lu et al. [10], it is necessary to investigate and define complexity from the different perspectives of the participants who are involved in the project. This paper analyzed complexity from the perspective of the contractor, and it determined the classification of the groups for the associated key elements of complexity. In this way, a basis for future research, which requires the establishment of a framework for evaluating the elements of complexity in contractors, was created. The creation of such a framework will provide the contractor with an adequate tool to analyze the impact of the project features on the activation of the risk and on the success of the project.

It is also important to emphasize that, for the different participants involved in the construction project, the complexity of the construction project needs to be defined at the different stages of the project. It is known that the construction project begins at different stages for different participants, but it is also necessary that each project participant define the degree of project complexity from their own perspective as soon as possible, immediately after he/she becomes involved in the project's implementation. From the investor perspective, complexity should be defined immediately, as soon as the initial

project realization planning starts—that is, in the project design phase. In the planning and design phase, it is mainly the designers and the supervising engineer who are involved in the project. At this stage of the project, it is necessary to determine the degree of project complexity from their perspectives, and thus influence the reduction in the overall complexity of the project by influencing the complexity related to the designer and the supervising engineer. Complexity from the perspective of the contractor—the subject of the research in this paper—should be analyzed upon the beginning of the contractor's involvement in the project. The contractor is included in the project during the execution stage. The inclusion of the contractor in the project also means the inclusion of other participants related to the execution of the work, i.e., subcontractors, suppliers of different resources, etc.

Although there have been a number of surveys of the complexity of construction projects over the last thirty years, surveys on the complexity of projects from the contractor's perspective are very rare. Today, given the methods of contracting, as well as the fact that the contractor employs the vast majority of the resources required for the project, the state of construction is such that most of a construction project's complexity is transferred to the contractor. Therefore, for the overall success of the construction project, it is necessary to analyze complexity as a significant determinant of the project from the perspective of the contractor. From the contractor's perspective, the project represents a subproject of the entire project, as seen from the investor's perspective. From the contractor's perspective, the success of the project can affect the success of the complete project. However, the success of the project from the contractor's perspective does not necessarily mean the success of the entire project, nor the opposite.

The present research is based on previous research and compares project complexity for the different participants involved in the construction project. In their 2012 research, Xia and Chan concluded that measuring project complexity is different for investors, designers, project managers, and contractors. In his research, Gidado [28] deals with project complexity in relation to time and money as the most important indicators (according to him) of how a construction project is managed by contractors. According to him, the situation of the contractor in relation to the other participants in the project is much more complex. The conclusion is that the degree of complexity to which the contractor is exposed is greater than the degree of complexity to which the other participants involved in the construction project are exposed. Brockmann and Girmscheid [46] state that contractors on large construction projects respond to the overall complexity, as well as the complexity of individual tasks by dividing them into smaller elements along their sections of functioning, and in this way, they can manage them more or less successfully. Contractors employ large amounts of resources, have less impact on the environment than investors, use state-of-the-art scientific and technological know-how, and combine different methods in the workflow. The contractor's situation is much more complex than the situation of the other participants involved in the construction project [28].

Taking into account the phases of construction projects, the largest number of interactions between the participants and the project elements occur in the project execution phase. At this stage, the project has the most participants, it correlates with the environment the most, and has the most financial flows present. Therefore, it is clear that this phase of the construction project is one in which the complexity of the project should receive special attention. All of this was confirmed by Winchur [72], who, through his research, came to the conclusion that the complexity of a construction project is the greatest at the stage in which the work is executed. The contractor largely controls the processes and has the most influence on the project at the work execution stage, and, therefore, the complexity of the project at this stage is of the greatest importance for him/her.

Information on project complexity can make it easier for contractors to make management decisions during the procurement process, to set project objectives and manage risks, and to determine project personnel [18]. When taking over project management from the contractor, each project manager's primary interest is the complexity of the project, which

is set for execution, and he/she seeks answers to the questions related to the characteristics that make up the complexity of the project.

Before defining the bid for work execution, contractors generally do not have adequate information about the project because they are only provided with data related to the estimated value of the work, the deadline for execution, and project documentation, and they are sometimes warned about special construction conditions through a tour of the construction site with the investor. Therefore, for a more successful project implementation, it would be very important that the contractor be able to determine the degree of project complexity from the available or possibly additional information based on the complexity model before defining the bid price. Thus, the contractor could influence and increase the chances his project has of success as a sub-project of the main project by looking at it from the investor's perspective. However, by increasing the success of performance in this way, the contractor can also greatly influence the increase in the investor's success in performance, as well as the success of all of the other participants involved in the execution of work on the construction project. Therefore, it is very important for the investor to provide the potential contractor with input data of the highest quality, which will then enable the contractor to determine the degree of project complexity as accurately as possible. Once the complexities of the construction project are better understood, it could enable the project management team to apply a proactive and front-end planning approach in the initiation phase in order to better manage scope changes in the delivery of the project, eventually improving the project performance [73]. Studies have shown that project complexity has an impact on the project performance but detailed studies on direct impacts are missing [11,74].

Traditional research related to project complexity focuses on components and elements of project complexity [21]. In accordance with all of the above, in order to define the classification of groups comprising the key elements associated with complexity from the perspective of the contractor, it is important to first analyze and systematize potential groups of complexity and elements of project complexity. The classification of groups and key elements of complexity for the contractor is the first step in analyzing the impact of a project's characteristics on the activation of risks and the success of the project.

*2.5. Review and Identification of Elements of Complexity from the Contractor's Perspective*

With the purpose of identifying the elements of complexity from the perspective of the contractor, a detailed review of the literature was conducted. The literature review was carried out in three steps. First, the search was performed based on the following keywords: project complexity, construction project complexity, complexity from contractor perspective, complexity, and project success. The first step of the search was performed in six databases, namely Science direct, ASCE Library, Taylor & Francis Online, Emerald insight, Academic Search Complete (EBSCO), and Google Scholar. The first step singled out 92 articles. The criterion for selecting the articles for the analysis was that the articles deal, either partly or completely, with the analysis of project complexity. In addition to searching the databases, in the second step, a search was performed in journals dealing with topics related to the field of research, namely: *Construction Management and Economics*, the *International Journal of Project Management*, the *Journal of Construction Engineering and Management*, the *Project Management Journal,* and the *Journal of Construction Engineering and Management*. The second step singled out an additional 34 articles. When searching the databases and journals, time filters were not included. In addition to the articles in the third step, two books and six PhD theses that were in the relevant fields of research were included.

By reviewing the literature in accordance with the above-stated methodology, a total of 37 articles defining the project complexity model were extracted. These 37 articles identified 267 different associated elements of complexity. For the purposes of the research, an analysis of these models was conducted. There is a large diversity of models of project complexity and complexity elements. This diversity can be illustrated by the variety of selected dimensions in the different models [75]. As stated, the key role in defining the

complexity of a project is the fact that it needs to be defined from the different perspectives of the participants involved in the project. Accordingly, when reviewing the literature and the elements of complexity, the existing elements of complexity and groups of complexity were structured to best suit the contractor's perceptions.

The literature review identified eight indicators that will be defined in the research as groups of project complexity based on the frequency of their occurrence in existing research. Through the reviewed literature, it was found that certain groups of complexity in certain studies also appear as elements of complexity in complexity models. When analyzing the frequency of occurrence of these complexity groups, these occurrences were also taken into account.

It is important to emphasize that when structuring groups of complexity and elements of complexity, different names related to the same characteristics of the project were linked. Thus, more credible data based on the frequency of occurrence of individual groups and elements of complexity as real characteristics of the project were obtained. The analysis of the research determined the frequency of occurrence for individual groups of complexity elements, which were determined to be components of the complexity framework from the contractor's perspective. All of this was determined on the basis of an analysis of previously conducted research, the applicability of parts of existing models, and interviews with construction project managers with years of experience. Reviewing the complexity groups with the goal of defining the framework identified eight complexity groups, namely the complexity of the project scope, organizational complexity, operational and technological complexity, the complexity of the project environment, complexity related to resources on the project, legal and socio-political complexity, and communication and economic complexity. Below is an overview of the frequency of their occurrence in previous surveys (Table 1).

**Table 1.** Project complexity groups—occurrence frequency in existing surveys.

| Complexity Groups | Frequency of Occurrence |
|---|---|
| Scope complexity | 75.67% |
| Organizational complexity | 59.46% |
| Operational and technological complexity | 56.76% |
| Environment complexity | 40.54% |
| Complexity related to resources on the project | 37.84% |
| Legal and socio political complexity | 27.03% |
| Communication complexity | 16.22% |
| Economic complexity | 18.92% |

After a review of the literature, the elements of complexity were classified into groups of complexities based on previous research. Some of the elements of complexity that related to the same project characteristics, but that had different names in the existing research, were merged into a unique element of complexity to obtain more credible results. After that, the analysis of the frequency at which the elements of complexity occurred in previous surveys began.

Since the previous surveys did not consider complexity from the perspective of the contractor, the need to add elements of complexity to certain groups of complexity and elements that were not represented in the previous surveys arose. These elements were considered to be a significant contributor to the adequate definition of the framework and the classification of key elements of complexity from the perspective of the contractor.

The review of existing research dealing with elements of complexity identified several elements of complexity that were allocated to groups of complexity by means of analysis. The complexity group related to operational and technological complexity includes 26 elements of complexity; scope complexity includes 29 elements; organizational complexity includes 31 elements; complexity related to resource use during the project includes 22 elements; legal and socio-political complexity includes 19 elements; economic complexity

includes 17 elements; communication complexity includes 17 elements; and environment complexity includes 21 elements of complexity. The remaining elements (36) could not be classified into one of the identified groups of project complexity from the contractor's perspective. The analysis of the frequency of occurrence of certain elements of complexity was then initiated. Based on these results as well as through the addition of additional elements of complexity, which are characteristic of the contractor, 15 elements of complexity were defined for each group of complexity of the project to be used in further research after conducting interviews with project managers with significant experience (Table 2).

**Table 2.** Project complexity groups with associated complexity elements.

| Project Complexity Groups with Associated Complexity Elements | Frequency of Occurrence | Project Complexity Groups with Associated Complexity Elements | Frequency of Occurrence |
|---|---|---|---|
| **OPERATIONAL AND TECHNOLOGICAL COMPLEXITY** | | **LEGAL AND SOCIO-POLITICAL COMPLEXITY** | |
| Incompleteness of the project documentation | 2.70% | Political impact of the project | 8.11% |
| Incorrect project documentation | 2.70% | Local legislation | 13.51% |
| Technological competence | 18.91% | Cultural diversity of participants | 13.51% |
| Technological diversity | 13.51% | Contract type | 2.70% |
| Usage of complex technologies | 16.22% | Culture of claims | 2.70% |
| Requirements of preparatory work | 5.40% | Investors on the project from a different country | 2.70% |
| Change of technology during the execution of works | 2.70% | Local experience | 5.40% |
| Presence of transport system near the construction site | 2.70% | Number of contracts | 2.70% |
| Energy Requirements | 5.40% | Changes in legislation during the execution of the project | 2.70% |
| Insufficient project data | 2.70% | Changing of the policies over the course of the project | 5.40% |
| Quality Requirements | 13.51% | Holding elections over the course of project execution | 2.70% |
| Inadequate bill of work expenses | 5.40% | Workforce fluctuations | 5.40% |
| Function of the structure being built | 2.70% | Preparedness of the local community for the project | 5.40% |
| Lack of quality management tools | 2.70% | Interest of the local community in the project | 5.40% |
| Technology that is unknown to the Investor | 10.81% | Political and social instabilities | 5.40% |
| **SCOPE COMPLEXITY** | | **ECONOMIC COMPLEXITY** | |
| Duration of the project | 27.02% | Project Financing | 13.51% |
| Project value | 16.22% | Change in prices in the course of the project | 8.11% |
| Number of activities in the project | 27.03% | Poor contractual price | 0.00% |
| Number of Critical Activities | 8.11% | Accuracy of the statistical office data, on the situation | 0.00% |
| Activity overlap | 29.73% | Funding from various sources | 5.40% |
| Overlap of Critical Activities | 8.11% | Currency of cost calculation | 5.40% |
| Overlap of the project phases | 8.11% | Availability of cost data for specific | 8.11% |
| Number of Cost Significant Items | 5.40% | Change in the investor's budget | 13.51% |
| Cost Significant Items on Critical Path | 0.00% | Economic stability of the investor | 10.81% |

**Table 2.** *Cont.*

| Project Complexity Groups with Associated Complexity Elements | Frequency of Occurrence | Project Complexity Groups with Associated Complexity Elements | Frequency of Occurrence |
|---|---|---|---|
| Interconnectedness of activities from different stages | 8.11% | Financial condition of the contractor | 5.40% |
| Changes in the scope of the project over the course of execution | 5.40% | Payment deadlines | 5.40% |
| Quantity of additional works | 8.11% | Number of variations in the project | 5.40% |
| Amount of activities with a long duration | 8.11% | Existence of a minimum chargeable amount | 2.70% |
| Size of the project in terms of funds | 8.11% | Changes in the global economy | 2.70% |
| Variety of project scope | 13.51% | The existence of advance payment | 0.00% |
| **ORGANIZATIONAL COMPLEXITY** | | **COMMUNICATION COMPLEXITY** | |
| Number of investors | 5.40% | Communication within the project team | 8.10% |
| Number of hierarchical levels in the project team | 10.81% | Communication between the project manager and the project team | 10.81% |
| Significance of the project for the parent organization | 8.11% | Relationship between the project manager and the parent organization | 8.11% |
| Number of construction site locations | 10.81% | Large amount of information on the project | 8.10% |
| Number of subcontractors | 5.40% | Communication with the supervising engineer | 0.00% |
| Number of suppliers | 16.22% | Communication with the investor | 5.40% |
| The influence of the supervising engineer | 10.81% | Procedures during the project | 16.21% |
| Subcontractor work on the critical path | 0.00% | Communication with subcontractors | 8.11% |
| Subcontractor work overlapping | 0.00% | Communication with suppliers | 5.40% |
| Size of the project team | 10.81% | Capacity of the project team to transfer the information | 5.40% |
| Multiple contractors on the project | 8.10% | Diversity of participant communication cultures | 8.11% |
| The importance of the project for the investor | 5.40% | Interdependence of the established procedures | 8.11% |
| Coordination of participants | 16.22% | Inconsistency of procedures | 5.40% |
| Changes of the project team members during the progress | 5.40% | Meetings | 0.00% |
| Interconnectedness of participants | 21.62% | Concealment of information between participants | 10.81% |
| **COMPLEXITY RELATED TO RESOURCES ON THE PROJECT** | | **ENVIRONMENT COMPLEXITY** | |
| Resource Quantity | 18.92% | Dependence on the environment | 10.81% |
| Diversity of material resources | 8.11% | Local climatic conditions | 13.51% |
| Diversity of the workforce | 5.40% | Geological conditions | 8.11% |
| Availability of material resources | 18.92% | Geographical location of the participants | 2.70% |
| Availability of a skilled workforce | 18.92% | Hydrological and hydrogeological conditions | 2.70% |
| Equipment availability | 10.81% | Stability of the environment | 5.40% |
| The variety in equipment | 10.81% | Extreme weather conditions | 5.40% |
| The experience of the project manager acquired on similar projects | 16.22% | Construction site in a public environment | 5.40% |
| Experience of the project team acquired on similar projects | 16.22% | Interaction between the technological system and the environment | 5.40% |

**Table 2.** *Cont.*

| Project Complexity Groups with Associated Complexity Elements | Frequency of Occurrence | Project Complexity Groups with Associated Complexity Elements | Frequency of Occurrence |
|---|---|---|---|
| Oscillations in the number of available human resources | 8.11% | Groundwater protection zone | 2.70% |
| Changes in the project manager in the course of execution | 21.62% | Construction site in contaminated environment | 2.70% |
| Smaller quantities of different material resources | 5.40% | Construction site in the historical core | 2.70% |
| Larger quantities of equal material resources | 5.40% | Incorrectly planned geological conditions | 2.70% |
| Resource Delivery | 8.11% | Construction site in traffic | 0.00% |
| Oscillations in the necessary equipment for the execution of works | 5.40% | Frequent variability in weather conditions | 8.11% |

When it comes to operational and technological complexity, it should be noted that in a significant number of surveys, operational and technological complexity are separated. Bearing in mind how connected they are at the work execution stage, in this case, they represent a single group of elements of complexity. The complexity of the project scope as a group of complexities appears, as stated above, most often in the existing research, from all groups of elements of complexity. This can be concluded if we consider that the two terms that are used have the same meaning, namely, the complexity of the project scope and the complexity of the task. For this complexity group, in addition to those determined by the literature review, the element of Cost Significant Items on the Critical Path has been included as an element characteristic of the contractor that can significantly affect the course of the project. As a group of complexities, organizational complexity was recognized at the very beginning of the process of defining and exploring the notion of complexity. Through analyzing the existing research, it can be concluded that there is virtually no research dealing with elements of complexity at any level without addressing a group or element of complexity related to the organization of the project. The contractor mainly uses the project resources during its implementation. Accordingly, the analysis of existing research indicates the existence of adequately defined elements of complexity for this complexity group by looking at them from the perspective of the contractor. The current state of the construction market results in resources, which can have a very large impact on its results from the perspective of the contractor. In the existing research, elements related to the quantity and diversity of resources, namely, labour and force, as well as elements related to the contractor's project manager, appear the most often. For this group of complexities based on research from existing research, 13 elements of complexity were distinguished based on the frequency of occurrence, while 2 other elements of complexity that are characteristic for the contractor were included, namely, Subcontractor Work on a Critical Path and Subcontractor Overlap. Legal and socio-political complexity was analyzed to a lesser extent in the existing research. This complexity represents an additional challenge for providing a quality definition for the associated elements of complexity. They were analyzed by looking at legislation and contracting on the project. These elements, at the work execution stage, have a greater impact on the implementation of the project from the contractor's perspective than from the investor's perspective. During the beginning of the realization of the project as well as when work on the project is being executed, the contractor can encounter various challenges related to the legislation but also related to the social aspects that surround the project being executed. The complexity group related to economic indicators was found in existing surveys the least, along with communication complexity. Nevertheless, the financial stability of the project is a key determinant of its implementation from any perspective. Any significant irregularity in this area may not only affect the success of the project but may also lead to its suspension or the total interruption of its implementation. The review of existing research dealing with elements of complexity

highlights Project Financing, as well as possible Budget Changes and Investor Economic Stability and Contractor Financial Status as the most important elements. Bearing in mind the insufficient processing of economic complexity and taking into account the economic complexity from the perspective of the contractor, the elements of complexity that will be used in further research are included in the elements of complexity related to the Poor Contract Price of Works, the accuracy of data of the Statistical Office on the state of the market, and the existence of advanced payment. Communication complexity plays a very important role in a complex system such as a construction project. As a complexity group, it is necessary for it, to be part of the complexity framework from the contractor's perspective. Communication complexity contains a large amount of information circulating about the project, as well as a large number of interactions with other participants involved in the project, along with the local community. Communication complexity, similar to economic complexity, is not significantly analyzed in the existing research. One of its forms can be found in 16% of the research that deals with elements of complexity in some way. Among the existing elements of complexity that are highlighted in the table below, elements related to the procedures on the project, as well as communication within the project team itself, are particularly prominent. As the communication complexity was not sufficiently analyzed in the existing research, and considering its importance for complexity from the perspective of the contractor, the inclusion of additional elements of complexity was performed, namely the inclusion of elements related to meetings, as well as communication with the supervising engineer. The environment of the project can influence its execution significantly, and, as a determinant of the project, has a great influence on the degree of complexity from the perspective of the contractor. In existing surveys of complexity, the environment of a project appeared in a significant number of surveys as one of the determinants of project implementation. By analyzing the frequency at which the elements related to the project environment occurred, as well as their applicability from the perspective of the contractor, 14 elements were singled out. The most common elements related to the environment were the Environmental Dependence of the project and the Local Climate Conditions of the project being executed. In addition to the fourteen elements singled out from the existing research, the element related to the Construction Site under traffic was included as an element of project complexity from the perspective of the contractor, which will be a part of the research for defining the model of the complexity for a specific construction project from the perspective of the contractor.

By examining the elements of complexity separated into groups of complexity, it can be established that for each complexity group, there are elements that stand out with regard to their presence in the existing research. The most common elements in the existing research are the overlap of activities, the project duration, and the number of activities on the project. All of these elements belong to the group of project scope complexity elements. In continuation of the present work based on the results of the pre-existing research, key elements of complexity for each complexity group are distinguished, and in this way, a classification of groups and the associated elements of complexity of the construction project was created from the perspective of the contractor. It is important to emphasize that the frequency of occurrence in the existing research has no impact on the present research or on the final definition of the key elements of complexity from the perspective of the contractor.

## 3. Research Methodology

The research presented in this paper was based on the need to determine how a project's characteristics affect the success of the construction project from the perspective of the contractor. Complexity characteristics do not all have the same impact on the success of a project [10,76], so it is important to understand and quantify the aggregated weight of each complexity element and its impact on the overall level of project complexity.

The theoretical framework of this research is explained in the previous section. In order to increase the performance of the project, the contractor should timely identify the

characteristics of the project that, as elements of complexity, affect the success of the project. The research consists of collecting data through a review of the literature made in the previous section, sending questionnaires and collecting answers to the questionnaire given by the project managers who work in organizations which are contractors on construction projects. After that, the analysis of the received data was performed, on the basis of which the results of the research will be presented. The analysis and the presentation of the research results were made on the basis of mathematical data processing and the application of the mean value of the obtained results.

To collect the necessary data to conduct this project, based on the determined input data, the focus group method was applied and consisted of 41 experts who provided answers to the questions asked. In order to determine the final form of the research, the research was first analyzed with five members of the focus group with the most experience as the contractor project managers. While defining the questionnaire in the initial interviews about the research itself, detailed interviews about the content of the questionnaire were conducted with five members of the focus group. Through initial discussions and an analysis of the prepared questionnaire, the number of complexity elements per group was reduced from 15 to 10.

The project managers considered that it was possible to reduce the number of complexity elements that would be addressed by the research immediately at the beginning of the survey. In addition, they considered that conducting extensive research with a large number of complexity elements with common characteristics would create a number of ambiguities when providing answers to the questions asked and would significantly affect the increase the scope of the overall research. Bearing in mind that the focus group consisted of representatives of contracting organizations with extensive experience, their suggestions were accepted. In this way, the key elements of complexity from the contractor's perspective had largely been identified already. The reduction in the number of complexity elements was mainly carried out by linking similar and easily connectable complexity elements into one common complexity element encompassing a slightly wider area based on the experiences of five members of the narrower focus group to conduct the research in a more efficient manner. In cases where linking alone was not sufficient to reduce the number of complexity elements to ten per group, the expulsion of those complexity elements whose appearance was less frequent compared to others was initiated by looking at the review of previous research, and some of the complexity elements that were found to be less suited to the perspective of the contractor were expelled through initial discussions with members of the focus group. In this way, the complexity groups containing other associated elements of complexity could be defined.

In the continuation of the research, it was necessary to determine the importance of the influence of individual elements of complexity on the degree of complexity of each complexity group and thus on the overall degree of complexity of the project. By determining the intensity of this impact, the classification of complexity groups and key elements of complexity from the perspective of the contractor could be determined.

In addition to the aforementioned five representatives of the focus group, the survey was sent to fifty-four more addresses, and we received answers from thirty-six of those addresses. A total of 18 respondents did not provide a response to the questionnaire. Of those eighteen non-responders, nine declared that the research presented here is a complex task that would take them a long time to complete and that they are currently unable to send a response. Three respondents stated that they were not sure if they could provide adequate answers to the above questions and that they would not submit their answers. No feedback was ever received from the remaining six addresses to which the questionnaires were sent. Nevertheless, the 41 responses that were received provided a significant sample that could adequately define the answers to the questions.

It is important to emphasize that more than half (53%) of the respondents who participated in the survey were working in at least two countries; this confirms them as having the necessary international experience, which can add relevance to the results of the sur-

vey (Table 3). It is possible that the diversity of the market could influence the different complexity elements and the overall measure of complexity. In addition, more than 80% of respondents had worked for two or more construction companies throughout their working life. This data indicates that the respondents have the necessary experience in different companies or systems to realistically assess the impact of the complexity of the project on the contractor without taking into account the characteristics of the system of only one organization. In addition, companies of different sizes—i.e., different number of workers—are represented in the survey. Based on this data, it is possible to analyze the results obtained by the research with regard to this indicator and to identify differences in the definition of key elements of complexity with regard to the size of the company.

**Table 3.** Characteristics of the responders.

| Characteristic | | Number of Respondents |
|---|---|---|
| Sex | Female | 37 |
| | Male | 4 |
| Age | <30 | 0 |
| | 30–40 | 28 |
| | 41–50 | 10 |
| | 51–60 | 2 |
| | >60 | 1 |
| Qualifications | NSS | 0 |
| | Secondary school | 0 |
| | University degree | 28 |
| | Master's degree in science | 12 |
| | Doctor of Science | 1 |
| Years of experience in managing exection of construction works | 5–10 | 21 |
| | 11–20 | 13 |
| | 21–30 | 4 |
| | More than 30 | 3 |
| Number of states in which the respondent worked | 1 | 20 |
| | 2 | 12 |
| | 3 | 4 |
| | 4 | 2 |
| | 5 and more | 3 |
| Number of the construction companies where the respondent worked | 1 | 8 |
| | 2 | 11 |
| | 3 | 9 |
| | 4 | 4 |
| | 5 and more | 9 |
| Number of workers in the respondents company | 0–50 | 4 |
| | 51–100 | 7 |
| | 101–250 | 5 |
| | 251–500 | 8 |
| | More than 30 | 17 |

## 4. Research Results

In the conducted research, the levels of importance of the individual elements of complexity for construction projects were defined from the perspectives of the contractors for each of the complexity groups based on the submitted answers. For the previously defined lists of complexity elements for each complexity group, the degree of importance of their impact on each group of complexity elements and on the overall complexity of the project was offered from the perspectives of the contractors. During the period of time in which the replies to the questionnaire were being sent back, there were no significant questions or ambiguities regarding the nature of the project, which represents a particular element of complexity that is part of the present research. The degree of importance was determined by the respondents by providing answers on a Likert scale of importance that ranged from 5 to 1. On the scale of importance determining the impact, 5 represents the largest possible impact, and number 1 represents the smallest possible impact that an element could have on the degree of complexity of the group. The impact with an intensity of 4 on the Likert scale represents a large impact on the degree of complexity of the group, 3 represents a medium impact, and 2 represents a low impact of elements of complexity on the degree of complexity of the group.

By analyzing the data obtained from the questionnaire responses and by creating a ranking of importance with regard to the mean value of the received responses, we were able to obtain the data presented below.

The elements are arranged based on level of importance with regard to the mean value of all 41 responses submitted over the course of the research.

When talking about the elements of operational and technological complexity, the elements that stand out are the ones related to how technology changes when work is being executed, as well as the incompleteness and inaccuracy of the project documentation. The least important elements of an operational and technological complexity group are Quality Control and Quality Requirements and Function of the structure being built (Table 4).

**Table 4.** Degrees of importance of elements of operational and technological complexity.

| Operational and Technological Complexity Group | Mean Value | Rankings |
|---|---|---|
| Change in technology intended for the execution of works during the execution of works | 3.9756 | 1. |
| Incomplete and inaccurate project documentation | 3.8537 | 2. |
| Technology intended for the execution of work | 3.7805 | 3. |
| Inadequate bill of work expenses | 3.7317 | 4. |
| Presence of transport system near the construction site | 3.3902 | 5. |
| Requirements of preparatory work | 2.7073 | 6. |
| Technology which is unknown to the Investor | 2.7073 | 7. |
| Energy Requirements | 2.6098 | 8. |
| Quality Control and Quality Requirements | 2.5366 | 9. |
| Function of the structure being built | 2.5366 | 10. |

Through the analysis of the results of the examination on the importance of individual elements of complexity in the group of complexities related to the scope of the project, it can again be concluded that the element with the greatest possibility of variability has the most significant impact on the project. Changes in the scope of the project during the period of time in which work is being conducted thus have a mean value of importance that amounts to 4.1220. The characteristics of the project that, according to the respondents, can play a significant role in the degree of complexity of this complexity group, are the number and the overlap of critical activities, as well as the number and the overlap of activities on the project. The overlap of project phases represents the least important element of complexity

out of all the isolated elements from the complexity group related to the scope of the project (Table 5).

**Table 5.** Degrees of importance of elements of scope complexity.

| Scope Complexity Group | Mean Value | Rankings |
|---|---|---|
| Changes in the scope of the project over the course of execution | 4.1220 | 1. |
| Number and overlap of Critical Activities | 4.0000 | 2. |
| Number and overlap of activities on the project | 3.8537 | 3. |
| Quantity of additional works | 3.7805 | 4. |
| Variety of project scope | 3.6585 | 5. |
| Number of Cost Significant Items and Cost Significant Items on a Critical Path | 3.5366 | 6. |
| Project value | 3.2927 | 7. |
| Size of the project in terms of funds | 3.2195 | 8. |
| Duration of the project | 3.1951 | 9. |
| Overlap of project phases and the interconnectedness of activities from different project phases | 3.1220 | 10. |

Organizational complexity as a determinant of complexity has been present from the very beginning of complexity theory. Regardless of what has been stated above, it does not necessarily represent the most important groups of complexities from the contractor's perspective.

The number of construction site locations itself, as well as the number of investors participating in the project, has the least important impact on the degree of complexity when talking about the organizational complexity of the project. The number of site locations and the number of investors represent the characteristics of the project that are known to the contractor since the contractor's inclusion in the project and their impact of those elements on the project can be properly planned without the significant changes being initiated while work is being completed. The least important elements of an organizational complexity group in accordance with survey results are the number of construction site locations and the number of investors (Table 6).

**Table 6.** Degrees of importance of elements of organizational complexity.

| Organizational Complexity Group | Mean Value | Rankings |
|---|---|---|
| The importance of the project for the investor | 4.0732 | 1. |
| The influence of the supervising engineer | 4.0244 | 2. |
| Significance of the project for the company | 3.8780 | 3. |
| Subcontractor works on the critical path | 3.8537 | 4. |
| Coordination of participants | 3.8049 | 5. |
| Multiple contractors on the project | 3.4878 | 6. |
| Number of subcontractors and suppliers | 3.1463 | 7. |
| Number of hierarchical levels in the project team | 3.0976 | 8. |
| Number of construction site locations | 2.8049 | 9. |
| Number of investors | 2.5854 | 10. |

Considering the problems in the construction market, it is expected that the research results show how the diversity of the workforce, as well as its availability, represent the most important element of complexity related to the resources required for the project. What can also be of great importance, in accordance with the results of the research, is the experience that the project manager has working on similar projects. The availability of such a resource significantly facilitates the position of contractors when work is being

executed on a project. The number of resources itself represents the least important element of this group of project complexity since it contains the least unknowns, so it can thus be planned and will not change significantly over the course of the project (Table 7).

**Table 7.** Degrees of importance of resource-related complexity elements.

| Complexity Group Related to Resources on the Project | Mean Value | Rankings |
|---|---|---|
| Diversity and Availability of Workforce | 4.1951 | 1. |
| The experience of the project manager acquired on similar projects | 4.1220 | 2. |
| Workforce fluctuations | 3.8049 | 3. |
| Experience of the project team acquired on similar projects | 3.7561 | 4. |
| Oscillations in the number of resources required on the project | 3.7317 | 5. |
| Change in the project manager over the course of execution | 3.5854 | 6. |
| Diversity and availability of material resources | 3.5122 | 7. |
| Diversity and availability of equipment | 3.4146 | 8. |
| Resource Delivery | 3.1707 | 9. |
| Resource Quantity | 3.0244 | 10. |

Bearing in mind that the execution of construction projects is an undertaking that drives the entire social community, the socio-political and legal elements surrounding the project play a significant role in its success. According to the research results, the political impact on the project is the most important element of the complexity of the project from this group. The least important elements of the legal and socio-political complexity group are the number of contracts and the cultural diversity of participants (Table 8).

**Table 8.** Degrees of importance of elements of socio-political complexity.

| Legal and Socio-Political Complexity Group | Mean Value | Rankings |
|---|---|---|
| Political impact on the project | 4.0732 | 1. |
| Local legislation | 4.0000 | 2. |
| Local experience | 3.9024 | 3. |
| Holding elections over the course of project execution | 3.8780 | 4. |
| Local community | 3.1951 | 5. |
| Changes in legislation during the execution of the project | 3.0488 | 6. |
| Contract type | 2.7561 | 7. |
| Culture of claims | 2.7073 | 8. |
| Number of contracts | 2.5854 | 9. |
| Cultural diversity of participants | 2.4634 | 10. |

Financing a project of any kind, including a construction project from the perspective of the contractor, represents a significant determinant of the complexity of the project's implementation. The most important elements of the economic complexity group are the financial condition of the contractor and the economic stability of the investor.

The existence of advanced payment and the currency of cost calculations represent the least important elements affecting the economic complexity of the project, especially when taking into account that they are known characteristics of the project from the very beginning and cannot change. What may partly affect the contractor is the currency of cost calculations, especially in certain volatile markets. However, this research mainly covers markets with stable currencies, and this research result is expected (Table 9).

Given its nature, communication complexity, which is mainly characterized by various types of uncertainties, represents a very significant group for the overall complexity of a project.

**Table 9.** Degrees of importance of elements of economic complexity.

| Economic Complexity Group | Mean Value | Rankings |
|---|---|---|
| Financial condition of the contractor | 4.2439 | 1. |
| Economic stability of the investor | 4.1463 | 2. |
| Project Financing | 4.0976 | 3. |
| Payment deadlines | 3.8780 | 4. |
| Number of variations on the project (impact of changes in the financial value of the order) | 3.5122 | 5. |
| Change in prices over the course of the project (adjustment for changes in | 3.3659 | 6. |
| Changes in the global economy | 3.2195 | 7. |
| Availability of cost data for specific activities | 3.0244 | 8. |
| The existence of advance payment | 2.8293 | 9. |
| Currency of cost calculation | 2.4878 | 10. |

From the contractor's point of view, in accordance with the results of the research, the most important element of complexity relates to communication with the supervising engineer and the investor, as well as concealing information between the participants in the project. When all of these elements of complexity have an adequate impact, they can significantly affect the overall complexity of the project and thus its results and success. For this complexity group, the least important element of complexity is the diversity of communication cultures of the participants involved in the project because the diversity of cultures cannot significantly contribute to the quality of communication itself if some other problem is not present (Table 10).

**Table 10.** Degrees of importance of elements of communication complexity of the project.

| Communication Complexity Group | Mean Value | Rankings |
|---|---|---|
| Communication with the supervising engineer and the investor | 4.1951 | 1. |
| Concealment of information between participants | 4.1707 | 2. |
| Relationship between the project manager and the parent organization | 4.0488 | 3. |
| Communication with subcontractors and suppliers | 4.0244 | 4. |
| Procedures on the project | 3.8537 | 5. |
| Communication within the project team | 3.7073 | 6. |
| Large amount of information about the project | 3.4390 | 7. |
| Capacity of the project team to transfer the information | 3.2927 | 8. |
| Meetings | 3.0488 | 9. |
| Diversity of participant communication cultures | 3.0244 | 10. |

The results obtained through the test indicate that the inaccuracy of the projected geological conditions, as well as the geological conditions themselves, have the most significant impact on the degree of complexity of this complexity group from the perspective of the contractor. However, the elements with the greatest share of uncertainty in their occurrence and impact on the execution of the project are shown to be the most significant elements of each complexity group. In accordance with the above, the elements that are known and whose impact can be predicted and that are related to the construction site in historical areas or in contaminated and groundwater protection zones have the least impact on the degree of complexity related to the environment of the project being executed. Via

proper planning, the contractor can significantly reduce their impact on the complexity and results of the project (Table 11).

**Table 11.** Degrees of importance of elements of complexity of the project environment.

| Environment Complexity Group | Mean Value | Rankings |
|---|---|---|
| Incorrectly planned geological conditions | 4.2927 | 1. |
| Geological conditions | 4.1951 | 2. |
| Construction site in traffic | 4.1220 | 3. |
| Local climatic conditions | 3.7805 | 4. |
| Interaction between the technological system and the environment | 3.7317 | 5. |
| Construction site in a public environment | 3.7073 | 6. |
| Hydrological and hydrogeological conditions | 3.5122 | 7. |
| Construction site in the historical core | 3.3902 | 8. |
| Construction site in contaminated environment | 3.1220 | 9. |
| Groundwater protection zone | 2.6585 | 10. |

By conducting the present research and by analyzing the results of the degree of influence of a particular element of complexity on a particular group affecting the project, the elements of complexity were ranked according to their level of importance. What is also important is that the research confirmed that the proposed elements of complexity, which were collected through the existing research—as well as from the suggestions of the focus group members with adequate experience in project management and in the execution of projects—formed appropriate elements of project complexity from the perspective of the contractor.

By distinguishing and agreeing on the importance of each element, a classification of complexity groups was created with the associated key elements of complexity from the perspective of the contractor. The research found that the elements of complexity that have uncertainty and dynamism as their basic characteristics have the greatest importance for the degree of complexity in individual groups. From the above, a clear link can be drawn to the previously stated claim that elements of complexity are the drivers of risk during a project.

The established framework consists of eight groups of complexity elements with ten associated complexity elements in each group (Table 12). The established classification represents the first significant result related to groups and elements of complexity from the contractor's perspective. As such, this classification is the basis for analyzing the impact of complexity elements on the activation of risks on the project and consequently on the success of the project from its perspective. Within this framework, the elements of complexity are classified with regard to their importance for the contractor. Depending on the contractor's needs for the project being executed, the contractor may also allocate a smaller number of key elements of complexity to possibly reduce the scope of the analysis of the impact of elements of complexity over the course of the project. Based on this classification and the conducted research, the classifications of key elements of complexity for different types of projects and the values of the projects that are being executed, as well as the types of contracts on the basis of which the construction work on the project is being executed, can also be defined.

Table 12. The classification of groups with associated key complexity elements from the contractor's perspective.

| Operational and Technological Complexity Group | Scope Complexity Group | Organizational Complexity Group | Complexity Group Related to Resources on the Project | Legal and Sociopolitical Complexity Group | Economic Complexity Group | Communication Complexity Group | Environment Complexity Group |
|---|---|---|---|---|---|---|---|
| Change in technology intended for the execution of works during the execution of works | Changes in the scope of the project over the course of execution | The importance of the project for the investor | Diversity and Availability of Workforce | Political impact on the project | Financial condition of the contractor | Communication with the supervising engineer and the investor | Incorrectly planned geological conditions |
| Incomplete and inaccurate project documentation | Number and overlap of Critical Activities | The influence of the supervising engineer | The experience of the project manager which is acquired on similar projects | Local legislation | Economic stability of the investor | Concealment of information between participants | Geological conditions |
| Technology intended for the execution of works | Number and overlap of activities on the project | Significance of the project for the company | Workforce fluctuation | Local experience | Project Financing | Relationship between the project manager and the parent organization | Construction site in traffic |
| Inadequate bill of work expenses | Quantity of additional work | Subcontractor works on the critical road | Experience of the project team acquired on similar projects | Holding elections in the course of project execution | Payment deadlines | Communication with subcontractors and suppliers | Local climatic conditions |
| Presence of transport system near the construction site | Variety of project scope | Coordination of participants | Oscillations in the number of resources required for the project | Local community | Number of variations on the project | Procedures on the project | Interaction between the technological system and the environment |
| Requirements of preparatory work | Number of Cost Significant Items and Cost Significant Items on a Critical Path | Multiple contractors on the project | Change in the project manager over the course of execution | Changes in legislation during the execution of the project | Change in prices over the course of the project | Communication within the project team | Construction site in a public environment |
| Technology unknown to the Investor | Project value | Number of subcontractors and suppliers | Diversity and availability of material resources | Contract type | Changes in the global economy | Large amount of information about the project | Hydrological and hydrogeological conditions |
| Energy Requirements | Size of the project in terms of funds | Number of hierarchical levels in the project team | Diversity and availability of equipment | Culture of claims | Availability of cost data for specific activities | Capacity of the project team to transfer the information | Construction site in the historical core |
| Quality Control and Quality Requirements | Duration of the project | Number of construction site locations | Resource Delivery | Number of contracts | The existence of advance payment | Meetings | Construction site in contaminated environment |
| Function of the structure being built | Overlap of project phases and the interconnectedness of activities from different project phases | Number of investors | Resource Quantity | Cultural diversity of participants | Currency of cost calculation | Diversity of participant communication cultures | Groundwater protection zone |

## 5. Discussion

In the existing research, the authors mainly established the fact that there are a significant number of defined complexity models with associated elements. The research found that the models of complexity that have been established for construction projects were mostly analyzed from the investor's perspective. Bearing in mind that the existing research found that complexity is different for the individual participants involved in the construction project, this research sought to classify groups of complexity with the associated key elements of complexity from the perspective of the contractor. Given the existing practices in construction, the contractor bears the greatest burden of the complexity of the project and the impact of the complexity of the project on the activation of risk and on the success of the construction project.

The defined classification of complexity consists, as mentioned before, of eight complexity groups with the associated 10 key elements of project complexity for each group.

In operational and technological complexity, there stand out elements of complexity which, as main characteristics, have a significant degree of uncertainty in regard to their appearance during the project being their basic characteristic. With all of the above in mind, the high level of importance that these elements of complexity in the group of operational and technological complexity have is quite logical. The function of the structure being built comes out as the least important element of complexity in a group of organizational and technological complexity. Viewed from the contractor's perspective, this element really cannot play a more significant role in the results of the contractor's project, especially when bearing in mind that for the contractor, operational and technological tasks arise from the project documentation and not from the function of the structure being built.

The increase in the number of activities, as well as the overlap of those activities during the execution of works, significantly increases the degree of complexity related to the scope of the project. This complexity element, as well as the element related to the increasing of a scope of works, can significantly affect the success of the project. Phase overlapping is something that is known to the contractor at the very beginning of the project, and it is a characteristic that he can affect with adequate planning and can reduce its influence on the results and the success of the project.

Through the discussions with the participants of construction projects, which were related to the problems on their projects, they mainly put the emphasis on organization problems. From the point of view of organizational complexity, the most important elements of complexity are the importance of the project for the investor and the influence of the supervising engineer on the organization of the work to be completed during the project. If the project, which is performed by the contractor, represents something crucial for the investor, which often means that it has to be completed before the agreed upon time, then the total complexity of the project in each of the segments increases. In addition, the supervising engineer, the person authorized to manage the project in accordance with his contract with the investor, can significantly influence the atmosphere during the period of time in which work is being executed and can, through his actions, have the most influence on the necessary time, as well as the costs of the contractor.

In today's market, which is characterized by a lack of work force and problems with deliveries and changes in the prices of material resources and equipment, the complexity related to resources plays an important role in the overall complexity of the project from the perspective of the contractor. Significant changes in resource prices are common, which can put the contractor in a situation in which the execution of works is not a profitable task. However, a thing that most significantly affects the area of research related to resources is the growing shortage in the workforce. The market is characterized by a lack of both a trained and educated workforce and workers without any education and experience. Therefore, in this market, there is an increasing tendency to find workforce from Eastern Europe, as well as from Asia, specifically Turkey, India, or Bangladesh. The adaptation of the market to these conditions will be time-consuming and will require significant investment.

The impact of politics depends significantly on the realization of individual construction projects, and it is therefore expected that the political impact plays a significant role on the project's complexity. Local legislation, given the nature of the performing organizations and their operations in different communities, also has a high impact on the success and degree of complexity of a project. If we look at larger construction projects, we can conclude that construction has long been globalized, and it is therefore expected that, if there is cultural diversity among the participants involved in the project, then it will not affect the results of contractors significantly.

The frequent increases in purchase prices that characterize today's market can create significant inconvenience for the contractor when work is being executed. If these price increases are reflected in several major projects that the contractor is carrying out, then it may lead to problems with the contractor's overall financial condition. The results of the research define how the financial condition of the contractor represents the most important element of complexity in this group. If the financial condition of the contractor is not in accordance with the needs of the project, then it is expected that it will cause an increase in the complexity of the execution of the project. In addition to the financial condition of the contractor, the financial condition of the investor is an almost equally important element of complexity. The financial condition of the investor determines the dynamics of payments and can consequently also affect the financial condition of the contractor; the importance of this element of complexity on the overall complexity of the project can be easily deduced from this fact. For the contractor, this can be particularly negative if the key investor, with whom the contractor may have several contracts with at a certain time, is experiencing financial problems. Such a scenario is highly negative for the contractor, and the contractor should therefore endeavour to avoid being dependent on only one or several key investors.

Quality communication at any level reduces the complexity the project and thus affects the increase in success and the decrease in the overall complexity of the project. If the communication during a project leads to distrust between the participants, the consequences can be extremely negative. Deficiencies in communication can occur within the contractor's project team itself, which the contractor can solve on their own within their organization after recognizing them. However, if appropriate communication with the other participants involved in the project is achieved during the project's execution, this can play a major role in simplifying its implementation, thus creating a positive atmosphere and trust and affecting the results and success of the project both from the perspective of the contractor and the overall success of the construction project.

The environment of the project represents an important determinant of its execution. A lack of or the inaccuracy of the data about the environment of the contractor's project will mislead the contractor, as they prepare for certain conditions and not knowing that they will be met with completely different conditions during the execution of the project. The contractor has to reorganize these changes as soon as possible in order to ensure that as little time is lost as possible. Such ventures can generate large amounts of additional costs, both for the contractor and, consequently, the investor.

Related to the existing frameworks of the complexity of construction projects, it is important to point out that the adopted framework of complexity from the perspective of the contractor has almost the same groups of complexity, such as the Nguyen model for Complexity of Transportation projects from 2015 [77]. This framework also confirms the importance of the elements of complexity that the authors have singled out in previous research. It was also confirmed that the elements of complexity are the same for all participants in the project, but do not have the same impact on each of the participants. A more significant comparison of the defined framework of complexity with similar frameworks cannot be made, given that the researchers in the previous period did not deal in more detail with the analysis of complexity from the perspective of contractors.

A defined classification of complexity groups with associated key complexity elements represents a good theoretical basis for the contractor, but it cannot introduce significant benefits for the contractor as a participant in the implementation of the construction project

without adequate quantification. Therefore, in future research, it will be necessary to establish a framework for evaluating the key elements of complexity with regard to their impact on the activation of risks and the success of the construction project. By establishing such a framework and putting it into practice, the contractor can influence the proper planning of the project and thus influence their success and overall business.

The results of this study should be viewed in light of several limitations. They are mainly related to the choice of respondents. Due to market constraints, the survey could not be conducted among project managers with more experience in different fields of construction projects. Caution is required in extending findings to construction companies of different dimensions, belonging to different fields of construction and with different organizational settings. The results of the research need to be analyzed with regard to the different characteristics of the companies from which the respondents come, and in this way to form new frameworks of complexity. Although the findings are based on data from several construction companies, mostly from Southeast Europe, these outcomes can still provide reference for other countries considering the parallel construction industry experiences.

In addition to the conducted research, in order to classify the key elements of complexity in more detail, the same elements can be analyzed with regard to the types of construction projects that contractors tend to execute the most, as well as the size of the contracting firms themselves and the size of the construction projects they execute. In this way, more precise key elements of complexity would be defined with regard to the different characteristics of contractors and their projects. Thus, certain key elements of complexity constitute a good basis for a more accurate definition of the results and could thus significantly affect contractor performance.

## 6. Conclusions

The aim of this article was to classify groups of complexity with the associated key elements of complexity from the perspective of the contractor working on a construction project. Existing research related to the complexity of projects and the complexity of construction projects were analyzed. By researching the existing literature in the field of complexity, it was established that the existing research rarely or does not deal with the complexity of construction projects from the perspective of the contractor. In addition, the complexity of construction projects was found to be different for the different participants involved in the project, and the key elements of complexity are different for each of them. Through the literature review, the groups of complexity and elements of complexity that appear in the existing research were distinguished. Based on the frequency of their occurrence and the experiences of members of the focus group, the elements of complexity that are characteristic of contractors were distinguished as input data to conduct the present research. The research conducted among the construction project managers resulted in the creation of a classification comprising eight groups and ten associated key elements of complexity for each group of complexity. In this way, a framework of complexity for construction projects was formed from the perspective of the contractor and will hopefully serve as a basis for contractors to quantify the impact of complexity on the success of the projects they perform. Bearing in mind the impact of the contractor on the construction project, determining the key elements of complexity from the contractor's perspective affects the overall success of construction projects. In future works, it will be necessary to analyze and quantify the impact of the key elements of complexity on the results of construction projects according to the contractor's impact on the initiation of risk during the project. Based on this, a framework will be created, and by applying it, the contractor will be able to properly plan the objectives of the projects being performed and will thus influence the success of those projects.

**Author Contributions:** Conceptualization, M.N. and A.C.; methodology, M.N. and A.C.; software, M.N. and A.C.; validation, M.N.; formal analysis, M.N.; investigation, M.N.; resources, M.N.; data curation, M.N.; writing—original draft preparation, M.N.; writing—review and editing, A.C. All authors have read and agreed to the published version of the manuscript.

**Funding:** This research received no external funding.

**Institutional Review Board Statement:** Not applicable.

**Informed Consent Statement:** Informed consent was obtained from all subjects involved in the study.

**Conflicts of Interest:** The authors declare no conflict of interest.

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
