# Peer review of "Classification of Key Elements of Construction Project Complexity from the Contractor Perspective"

_buildings, doi:10.3390/buildings12050696_

Round 1

Reviewer 1 Report

Thank you for the opportunity to review the manuscript "Classification of Key Elements of Construction Project Complexity from the Contractor Perspective". The Authors classified groups of complexity with the associated key elements of complexity from the perspective of the contractor working on a construction project. I think the manuscript is well written and presents interesting data. However, the manuscript might fit better in a journal that has a more particular focus on this issue. In order to improve the quality of the manuscript, some comments from this reviewer can be seen as follows:

1- The problem statement and the focus of research is not well defined in the abstract, therefore, this reviewer needs to be reorganised.

2- The introduction section seems scattered; for instance, the authors mentioned the objectives of the research in the third paragraph. However, the introduction section should be narrow down towards the objectives and scope of the research at the end. Thus, this reviewer recommends rewriting to improve the readability of the manuscript.  

3- This reviewer appreciates the reviwe of the literature done by the authors, however, it would be better to have sub-sections under the 2.1 and 2.2. 

4- The Discussion section is very conclusive. This reviewer suggest to reduce the Findings and Result section, and confine it to the specific results only (to avoid any redundancy). By this way forward, the authors can discuss more the results in the "Discussion section". 

5- This reviewer also encourge the authors to enlist the contribution and findings in the bullet points in the conclusion section.

I appreciate the idea and the theortical work done by the author. I hope my comments will help further advance the manuscript.

Reviewer 2 Report

The authors have conducted a research on complexity in construction projects. Although the topic is interesting, major improvements are needed before it can be published in Buildings. My comments are as follows.

  • Is this manuscript submitted to “Sustainability” or “Buildings”? the heading and the journal to which the manuscript is submitted are not consistent.
  • Most of the references are very old and the reviewer can see only one reference after 2022. This makes the manuscript unsuitable for publication as the gap in research and the need for conducting this research is not justified while an extensive review of literature is not carried out by the authors. For instance:

“Although research related to the complexity of construction projects has intensified in the last twenty years, the review of the literature found that there are no significant complexity studies that focus on the perspectives of contractors.”

How did the authors conclude the above sentence without reviewing recent literature on this topic?

  • The research questions/ aim and objectives must be clearly mentioned in the last paragraph of the introduction.
  • Section 3, is part of the literature review and should be considered as a sub-section (2.3).
  • The first paragraph of section 4 is too long and it should be split into 2-3 paragraphs.
  • A research framework should be added and the software used for the analysis and the test for validation etc. should be mentioned clearly.
  • Figure 1 is redundant. It can be mentioned in the text in the methodology section, as it is just a simple Likert scale.
  • In the discussion section, did the authors compare the findings of their research with similar research? This section should be improved significantly based on the review of recently published papers in this field.
  • What are the limitations of the research?
